# Efficient Optimization for Linear Dynamical Systems with Applications to Clustering and Sparse Coding

**Wenbing Huang**[1,3], **Mehrtash Harandi**[2], **Tong Zhang**[2]

**Lijie Fan**[3], **Fuchun Sun**[3], **Junzhou Huang**[1]

[1] Tencent AI Lab. ;
[2] Data61, CSIRO and Australian National University, Australia;
[3] Department of Computer Science and Technology, Tsinghua University,
Tsinghua National Lab. for Information Science and Technology (TNList);
[1]{helendhuang, joehhuang}@tencent.com
[2]{mehrtash.harandi@data61.csiro.au, tong.zhang@anu.edu.cn}
[3]{flj14@mails, fcsun@mail}.tsinghua.edu.cn

## Abstract

Linear Dynamical Systems (LDSs) are fundamental tools for modeling spatio-temporal data in various disciplines. Though rich in modeling, analyzing LDSs is not free of difficulty, mainly because LDSs do not comply with Euclidean geometry and hence conventional learning techniques can not be applied directly. In this paper, we propose an efficient projected gradient descent method to minimize a general form of a loss function and demonstrate how clustering and sparse coding with LDSs can be solved by the proposed method efficiently. To this end, we first derive a novel canonical form for representing the parameters of an LDS, and then show how gradient-descent updates through the projection on the space of LDSs can be achieved dexterously. In contrast to previous studies, our solution avoids any approximation in LDS modeling or during the optimization process. Extensive experiments reveal the superior performance of the proposed method in terms of the convergence and classification accuracy over state-of-the-art techniques.

## 1 Introduction

Learning from spatio-temporal data is an active research area in computer vision, signal processing and robotics. Examples include dynamic texture classification [1], video action recognition [2, 3, 4] and robotic tactile sensing [5]. One kind of the popular models for analyzing spatio-temporal data is Linear Dynamical Systems (LDSs) [1]. Specifically, LDSs apply parametric equations to model the spatio-temporal data. The optimal system parameters learned from the input are employed as the descriptor of each spatio-temporal sequence. The benefits of applying LDSs are two-fold: **1.** LDSs are generative models and their parameters are learned in an unsupervised manner. This makes LDSs suitable choices for not only classification but also interpolation/extrapolation/generation of spatio-temporal sequences [1, 6, 7]; **2.** Unlike vectorial ARMA models [8], LDSs are less prone to the curse of dimensionality as a result of their lower-dimensional state space [9].

Clustering [10] and coding [5] LDSs are two fundamental problems that motivate this work. The clustering task is to group LDS models based on some given similarity metrics. The problem of coding, especially sparse coding, is to identify a dictionary of LDSs along their associated sparse codes to best reconstruct a collection of LDSs. Given a set of LDSs, the key problems of clustering and sparse coding are computing the mean and finding the LDS atoms, respectively, both of which are **not** easy tasks by any measure. Due to an infinite number of equivalent transformations for

the system parameters [1], the space of LDSs is non-Euclidean. This in turn makes the direct use of traditional techniques (*e.g.*, conventional sparse solvers) inapplicable. To get around the difficulties induced by the non-Euclidean geometry, previous studies (*e.g.*, [11, 12, 13, 5]) resort to various approximations, either in modeling or during optimization. For instance, the authors in [11] approximated the clustering mean by finding the closest sample under a certain embedding. As we will see in our experiments, involving approximations into the solutions exhibits inevitable limitations to the algorithmic performance.

This paper develops a gradient-based method to solve the clustering and sparse coding tasks efficiently without any approximation involved. To this end, we reformulate the optimization problems for these two different tasks and then unify them into one common problem by making use of the kernel trick. However, there exist several challenges to address this common problem efficiently. The first challenge comes from the aforementioned invariance property on the LDS parameters. To attack this challenge, we introduce a novel canonical form of the system parameters that is insensitive to the equivalent changes. The second challenge comes from the fact that the optimization problem of interest requires solving Discrete Lyapunov Equations (DLEs). At first glance, such a dependency makes backpropagating the gradients through DLEs more complicated. Interestingly, we prove that the gradients can be exactly derived by solving another DLE in the end, which makes our optimization much simpler and more efficient. Finally, as suggested by [14], the LDS parameters, *i.e.*, the transition and measurement matrices require to be stable and orthogonal, respectively. Under our canonical representation, the stability constraint is reduced to the bound constraint. We then make use of the Cayley-transformation [15] to maintain orthogonality and perform the bound-normalization to accomplish stability. Clustering and sparse coding can be combined with high-level pooling frameworks (*e.g.*, bag-of-systems [11] and spatial-temporal-pyramid-matching [16]) for classifying dynamic textures. Our experiments on such kind of data demonstrate that the proposed methods outperform state-of-the-art techniques in terms of the convergence and classification accuracy.

## 2   Related Work

**LDS modeling.** In the literature, various non-Euclidean metrics have been proposed to measure the distances between LDSs, such as Kullback-Leibler divergence [17], Chernoff distance [18], Binet-Cauchy kernel [19] and group distance [14]. This paper follows the works in [20, 21, 11, 12] to represent an LDS by making use of the *extended observability subspace*; comparing LDSs is then achieved by measuring the *subspace angles* [22].

**Clustering LDSs.** In its simplest form, clustering LDSs can be achieved by alternating between two sub-processes: 1) assigning LDSs to the closest clusters using a similarity measure; 2) computing the mean of the LDSs within the same cluster. However, as the space of LDSs is non-Euclidean, computing means on this space is not straightforward. In [12], the authors embedded LDSs into a finite Grassmann manifold by representing each LDS with its finite observability subspace and then cluster LDSs on that manifold. In contrast, our method applies the extended observability subspace to represent LDSs. In this way, not only the fully temporal evolution of the input sequence is taken into account, but also and as will be shown shortly, the computational cost is reduced. The solution proposed by [11] also represent LDSs with extended observability subspaces; but it approximates the mean by finding a sample that is closest to the mean using the concept of Multidimensional Scaling (MDS). Instead, our method finds the system tuple of the exact mean for the given group of LDSs without relying on any approximation. Afsari *et al*. [14] cluster LDSs by first aligning the parameters of LDSs in their equivalence space. However, the method of Afsari *et al*. is agnostic to the joint behavior of transition and measurement matrices and treat them independently. Other related studies include probabilistic framework for clustering LDSs [23, 24].

**Sparse Coding with LDSs.** Combining sparse coding with LDS modeling could further promote the classification performance [13]. However, similar to the clustering task, the non-Euclidean structure makes it hard to formulate the reconstruction objective and update the dictionary atoms on the space of LDSs. To address this issue, [13] embedded LDSs into the space of symmetric matrices by representing each LDS with its finite observability subspace. With this embedding, dictionary learning can be performed in the Euclidean space. In [5], the authors employ the extended observability subspaces as the LDS descriptors; however, to update the dictionary, the authors enforce symmetric constraints on the the transition matrices. Different from previous studies, our model

works on the the original LDS model and does not enforce any additional constraint to the transition matrices.

To sum up, in contrast to previous studies [12, 11, 14, 13, 5], this paper solves the clustering and sparse coding problems in a novel way regarding the following aspects. First, we unify the optimizing objective functions for both clustering and sparse coding; Second, we avoid any additional constraints (*e.g.* symmetric transition in [5] and finite observability in [12, 13]) for the solution; Finally, we propose a canonical formulation of the LDS tuple to facilitate the optimization.

## 3 LDS Modeling

LDSs describe time series through the following model [1]:

$$\begin{cases} \boldsymbol{y}(t) & = \overline{\boldsymbol{y}} + \boldsymbol{C}\boldsymbol{x}(t) + \boldsymbol{w}(t) \\ \boldsymbol{x}(t+1) & = \boldsymbol{A}\boldsymbol{x}(t) + \boldsymbol{B}\boldsymbol{v}(t), \end{cases} \tag{1}$$

with $\mathbb{R}^{m \times \tau} \ni \boldsymbol{Y} = [\boldsymbol{y}(1), \cdots, \boldsymbol{y}(\tau)]$ and $\mathbb{R}^{n \times \tau} \ni \boldsymbol{X} = [\boldsymbol{x}(1), \cdots, \boldsymbol{x}(\tau)]$ representing the observed variables and the hidden states of the system, respectively. Furthermore, $\overline{\boldsymbol{y}} \in \mathbb{R}^m$ is the mean of $\boldsymbol{Y}$; $\boldsymbol{A} \in \mathbb{R}^{n \times n}$ is the transition matrix of the model; $\boldsymbol{B} \in \mathbb{R}^{n \times n_v}$ ($n_v \leq n$) is the noise transformation matrix; $\boldsymbol{C} \in \mathbb{R}^{m \times n}$ is the measurement matrix; $\boldsymbol{v}(t) \sim \mathcal{N}(0, \mathbf{I}_{n_v})$ and $\boldsymbol{w}(t) \sim \mathcal{N}(0, \boldsymbol{\Omega})$ denoting the process and measurement noise components, respectively. We also assume that $n \ll m$ and $\boldsymbol{C}$ has full rank. Overall, generating the observed variables is governed by the parameters $\Theta = \{\boldsymbol{x}(1), \overline{\boldsymbol{y}}, \boldsymbol{A}, \boldsymbol{B}, \boldsymbol{C}, \boldsymbol{\Omega}\}$.

**System Identification.** The system parameters $\boldsymbol{A}$ and $\boldsymbol{C}$ of Eq. (1) describe the dynamics and spatial patterns of the input sequence, respectively [11]. Therefore, the tuple $(\boldsymbol{A}, \boldsymbol{C})$ is a desired descriptor for spatio-temporal data. Finding the optimal tuple $(\boldsymbol{A}, \boldsymbol{C})$ is known as *system identification*. A popular and efficient method for *system identification* is proposed in [1]. This method requires the columns of $\boldsymbol{C}$ to be orthogonal, *i.e.*, $\boldsymbol{C}$ is a point on the *Stiefel* manifold defined as $\text{ST}(m, n) = \{\boldsymbol{C} \in \mathbb{R}^{m \times n} | \boldsymbol{C}^{\text{T}}\boldsymbol{C} = \mathbf{I}_n\}$. The transition matrix $\boldsymbol{A}$ obtained by the method of [1] is not naturally stable. An LDS is stable if its *spectral radius*, *i.e.* the maximum eigenvalue of its transition matrix denoted by $\rho(\boldsymbol{A})$ is less than one. To obtain a stable transition matrix, [5] propose a soft-normalization technique which is our choice in this paper. Therefore, we are interested in the LDS tuple with the constraints,

$$\mathcal{C} = \{\boldsymbol{C}^{\text{T}}\boldsymbol{C} = \mathbf{I}_n, \rho(\boldsymbol{A}) < 1\}. \tag{2}$$

**Equivalent Representation.** Studying Eq. (1) shows that the output of the system remains unchanged under linear transformations of the state basis [1]. More specifically, an LDS has an equivalent class of representations, *i.e.*,

$$(\boldsymbol{A}, \boldsymbol{C}) \quad \sim \quad (\boldsymbol{P}^T \boldsymbol{A} \boldsymbol{P}, \boldsymbol{C} \boldsymbol{P}) \tag{3}$$

for any $\boldsymbol{P} \in \mathcal{O}(n)$[1]. For simplicity, the equivalence in Eq.(3) is called as *P-equivalence*.

Obviously comparing LDSs through Euclidean distance between the associated tuples is inaccurate as a result of *P-equivalence*. To circumvent this difficulty, a family of approaches apply the extended observability subspace to represent an LDS [20, 21, 11, 5]. Below, we briefly review this topic.

**Extended Observability Subspace.** The expected output sequence of Eq. (1) [12] is calculated as

$$[E[\boldsymbol{y}(1)]; E[\boldsymbol{y}(2)]; E[\boldsymbol{y}(3)]; \cdots] = [\boldsymbol{C}; \boldsymbol{C}\boldsymbol{A}; \boldsymbol{C}\boldsymbol{A}^2; \cdots]\boldsymbol{x}(1) = \boldsymbol{O}_\infty(\boldsymbol{A}, \boldsymbol{C})\boldsymbol{x}(1), \tag{4}$$

where $\boldsymbol{O}_\infty(\boldsymbol{A}, \boldsymbol{C}) \in \mathcal{R}^{\infty \times n}$ is called as the *extended observability matrix* of the LDS associated to $(\boldsymbol{A}, \boldsymbol{C})$. Let $\boldsymbol{S}(\boldsymbol{A}, \boldsymbol{C})$ denote the *extended observability subspace* spanned by the columns of $\boldsymbol{O}_\infty(\boldsymbol{A}, \boldsymbol{C})$. Obviously, the extended observability subspace is invariant to P-equivalence, *i.e.*, $\boldsymbol{S}(\boldsymbol{A}, \boldsymbol{C}) = \boldsymbol{S}(\boldsymbol{P}^{\text{T}}\boldsymbol{A}\boldsymbol{P}, \boldsymbol{C}\boldsymbol{P})$. In addition, the extended observability subspace is capable of containing the fully temporal evolution of the input sequence as observed from Eq. (4).

## 4 Our Approach

In this section, we first unify the optimizations for clustering and sparse coding with LDSs by making use of the kernel functions. Next, we present our method to address this optimization problem.

## 4.1 Problem Formulation

We recall that each LDS is represented by its extended observability subspace. Clustering or sparse coding in the space of extended observability subspaces is not straightforward because the underlying geometry is non-Euclidean. Our idea here is to implicitly map the subspaces to a Reproducing Kernel Hilbert Space (RKHS). For better readability, we simplify the subspace induced by $S(A_i, C_i)$ as $S_i$ in the rest of this section if no ambiguity is caused. We denote the implicit mapping defined by a positive definite kernel $k(S_1, S_2) = \phi(S_1)^T \phi(S_2)$ as $\phi : S \mapsto \mathcal{H}$. Various kernels [25, 19, 5] based on extended observability subspaces have been proposed to measure the similarity between LDSs. Though the proposed method is general in nature, in the rest of the paper we employ the projection kernel [5] due to its simplicity. The projection kernel is defined as

$$k_p(S_1, S_2) = \mathrm{Tr}(G_{11}^{-1} G_{12} G_{22}^{-1} G_{21}), \tag{5}$$

where $\mathrm{Tr}(\cdot)$ computes the trace and the product matrices $G_{ij} = O_\infty^T(A_i, C_i) O_\infty(A_j, C_j) = \sum_{t=0}^\infty (A_i^T)^t C_i^T C_j A_j^t$, for $i, j \in \{1, 2\}$ are obtained by solving the following DLE

$$A_i^T G_{ij} A_j - G_{ij} = -C_i^T C_j. \tag{6}$$

The solution of DLE exists and is unique when both $A_i$ and $A_j$ are stable [22]. DLE can be solved by a numerical algorithm with the computational complexity of $O(n^3)$ [26], where $n$ is the hidden dimension and is usually very small (see Eq. (1)).

**Clustering.** As discussed before, the key of clustering is to compute the mean for the given set of LDSs. While several works [12, 11, 14] have been developed for computing the mean, none of their solutions are derived in the kernel form. The mean defined by the implicit mapping is

$$\min_{A_m, C_m} \frac{1}{N} \sum_i^N \|\phi(S_m) - \phi(S_i)\|^2 \quad \text{s.t.} \ (A_m, C_m) \in \mathcal{C}, \tag{7}$$

where $S_m$ is the mean subspace and $S_i$ are data subspaces. Removing the terms that are independent from $S_m$ (e.g., $\phi(S_m)^T \phi(S_m) = 1$) leads to

$$\min_{A_m, C_m} -\frac{2}{N} \sum_i^N k(S_m, S_i) \quad \text{s.t.} \ (A_m, C_m) \in \mathcal{C}. \tag{8}$$

**Sparse Coding.** The problem of sparse coding in the RKHS is written as [13]

$$\min_{\{A'_j, C'_j\}_{j=1}^J} \frac{1}{N} \sum_i^N \|\phi(S_i) - \sum_{j=1}^J z_{i,j} \phi(S'_j)\|^2 + \lambda \|z_i\|_1, \quad \text{s.t.} \ (A'_j, C'_j) \in \mathcal{C}, j = 1, \cdots, J; \tag{9}$$

where $\{S_i\}_{i=1}^N$ are the data subspaces; $\{S'_j\}_{j=1}^J$ are the dictionary subspaces; $z_{i,j}$ is the sparse code of data $S_i$ over atom $S'_j$; $\mathcal{R}^J \in z_i = [z_{i,1}; \cdots; z_{i,J}]$ and $\lambda$ is the sparsity factor. Eq. (9) shares the same form as those in [13, 5]; however, here we apply the extended observability subspaces and perform no additional constraint on the transition matrices.

To perform sparse coding, we alternative between the two phases: 1) computing the sparse codes given LDS dictionary, which is similar to the conventional sparse coding task [13]; 2) optimizing each dictionary atom with the codes fixed. Specifically, updating the $r$-th atom with other atoms fixed gives the kernel formulation of the objective as

$$\Gamma_r = \frac{1}{N} \sum_i^N -z_{i,r} k(S'_r, S_i) + \sum_{j=1, j \neq r}^J z_{i,r} z_{i,j} k(S'_r, S'_j). \tag{10}$$

**Common Problem.** Clearly, Eq. (8) and (10) have the common form as

$$\min_{A, C} \frac{1}{N} \sum_{i=1}^N \beta_i k(S(A, C), S(A_i, C_i)) \quad \text{s.t.} \ (A, C) \in \mathcal{C}. \tag{11}$$

Here, $(A, C)$ is the LDS tuple to be identified; $\{(A_i, C_i)\}_{i=1}^N$ are given LDSs; $\{\beta_i\}_{i=1}^N$ are the task-dependent coefficients (are specified in Eq. (8) and Eq. (10)).

To minimize (11), we resort to the Projected Gradient Descent (PGD) method. Note that the solution space in (11) is redundant due to the invariance induced by P-equivalence (Eq. (3)). We thus devise a canonical representation of the system tuple (see Theorem 1). The canonical form not only confines the search space but also simplifies the stability constraint to a bound constraint. We then compute the gradients with respect to the system tuple by backpropagating the gradients through DLEs (see Theorem 4). Finally, we project the gradients to feasible regions of the system tuples via Caylay-transformation (Eq. (16-17)) and bound-normalization (Eq. (18)). We now present the details.

## 4.2 Canonical Representation

**Theorem 1.** *For any given LDS, the system tuple $(\boldsymbol{A}, \boldsymbol{C}) \in \mathbb{R}^{n \times n} \times \mathbb{R}^{m \times n}$ and all its equivalent representations have the canonical form $(\boldsymbol{\Lambda V}, \boldsymbol{U})$, where $\boldsymbol{U} \in \mathrm{ST}(m, n)$, $\boldsymbol{V} \in \mathcal{O}(n)$ and $\boldsymbol{\Lambda} \in \mathbb{R}^{n \times n}$ is diagonal with the diagonal elements arranged in a descend order, i.e. $\lambda_1 \geq \lambda_2 \geq \cdots \geq \lambda_n$ [2].*

**Remark 2.** *The proof of Theorem 1 (presented in the supplementary material) requires the SVD decomposition that is not necessarily unique [27], thus the canonical form of a system tuple is not unique. Even so, the free dimensionality of the canonical space (i.e., $mn$) is less than that of the original tuples (i.e., $mn + \frac{n(n-1)}{2}$) within the feasible region of $\mathcal{C}$. This is due to the invariance induced by P-equivalence (Eq. (3)) if one optimizes (11) in the original form of the system tuple.*

**Remark 3.** *It is easy to see that the stability (i.e., $\rho(\boldsymbol{A}) < 1$) translates into the constraint $|\lambda_i| < 1$ in the canonical representation with $\lambda_i$ being the $i$-th diagonal element of $\boldsymbol{\Lambda}$. As such, problem (11) can be cast as*

$$\min_{\boldsymbol{\Lambda}, \boldsymbol{V}, \boldsymbol{U}} \frac{1}{N} \sum_{i=1}^{N} \beta_i k(\boldsymbol{S}(\boldsymbol{\Lambda V}, \boldsymbol{U}), \boldsymbol{S}(\boldsymbol{A}_i, \boldsymbol{C}_i)),$$
$$\text{s.t. } \boldsymbol{V}^{\mathrm{T}} \boldsymbol{V} = \mathbf{I}_n; \ \boldsymbol{U}^{\mathrm{T}} \boldsymbol{U} = \mathbf{I}_n; \ |\lambda_i| < 1, i = 1, \cdots, n. \tag{12}$$

*A feasible solution of (11) can be obtained by minimizing (12) and the stability constraint in (11) is reduced to a bound constraint in (12).*

The canonical form derived from Theorem 1 is central to our methods. It is because with the canonical form, we can simplify the stability constraint to a bound one, thus making the solution simpler and more efficient. We note that even with conditions on one single LDS, optimizing the original form of A with the stability constraint is tedious (*e.g.*, [7] and we note that the tasks addressed in our paper are more complicated where far more than one LDS are required to optimize). Furthermore, the canonical form enables us to reduce the redundancy of the LDS tuple (see Remark 3). To be specific, with canonical form, one needs to update only $n$ singular values rather than the entire $\boldsymbol{A}$ matrix. Also optimization with the canonical representations avoids numerical instabilities related to equivalent classes, thus facilitating the optimization.

## 4.3 Passing Gradients Through DLEs

According to the definition of the projection kernel, to obtain $k(\boldsymbol{S}(\boldsymbol{A}, \boldsymbol{C}), \boldsymbol{S}(\boldsymbol{A}_i, \boldsymbol{C}_i))$ for (11) (note that in the canonical form $\boldsymbol{A} = \boldsymbol{\Lambda V}$ and $\boldsymbol{C} = \boldsymbol{U}$), computing the product-matrices $\boldsymbol{G}_i = \sum_{t=0}^{\infty} (\boldsymbol{A}^{\mathrm{T}})^t \boldsymbol{C}^{\mathrm{T}} \boldsymbol{C}_i \boldsymbol{A}_i^t$ are required. To compute the gradients of the objective in (11) shown by $\Gamma$ w.r.t. the tuple $\boldsymbol{\Theta} = (\boldsymbol{A}, \boldsymbol{C})$, we make use of the chain rule in the vectorized form as

$$\frac{\partial \Gamma}{\partial \boldsymbol{\Theta}:} = \sum_i \frac{\partial \Gamma}{\partial \boldsymbol{G}_i:} \frac{\partial \boldsymbol{G}_i:}{\partial \boldsymbol{\Theta}:}. \tag{13}$$

While computing $\frac{\partial \Gamma}{\partial \boldsymbol{G}_i:}$ is straightforward, deriving $\frac{\partial \boldsymbol{G}_i:}{\partial \boldsymbol{\Theta}:}$ is non-trivial as the values of the product-matrices $\boldsymbol{G}_i$ are obtained by an infinite summation. The following theorem proves that the gradients are derived by solving an induced DLE.

**Theorem 4.** *Let the extended observability matrices of two LDSs $(\boldsymbol{A}_1, \boldsymbol{C}_1)$ and $(\boldsymbol{A}_2, \boldsymbol{C}_2)$ be $\boldsymbol{O}_1$ and $\boldsymbol{O}_2$, respectively. Furthermore, let $\boldsymbol{G}_{12} = \boldsymbol{O}_1^{\mathrm{T}} \boldsymbol{O}_2 = \sum_{t=0}^{\infty} (\boldsymbol{A}_1^{\mathrm{T}})^t \boldsymbol{C}_1^{\mathrm{T}} \boldsymbol{C}_2 \boldsymbol{A}_2^t$ be the product-matrix between $\boldsymbol{O}_1$ and $\boldsymbol{O}_2$. Given the gradient of the objective function with respect to the product-matrix $\frac{\partial \Gamma}{\partial \boldsymbol{G}_{12}} \doteq \boldsymbol{H}$, the gradients with respect to the system parameters are*

$$\frac{\partial \Gamma}{\partial \boldsymbol{A}_1} = \boldsymbol{G}_{12} \boldsymbol{A}_2 \boldsymbol{R}_{12}^{\mathrm{T}}, \quad \frac{\partial \Gamma}{\partial \boldsymbol{C}_1} = \boldsymbol{C}_2 \boldsymbol{R}_{12}^{\mathrm{T}}, \quad \frac{\partial \Gamma}{\partial \boldsymbol{A}_2} = \boldsymbol{G}_{12}^{\mathrm{T}} \boldsymbol{A}_1 \boldsymbol{R}_{12}, \quad \frac{\partial \Gamma}{\partial \boldsymbol{C}_2} = \boldsymbol{C}_1 \boldsymbol{R}_{12}, \tag{14}$$

*where $\boldsymbol{R}_{12}$ is obtained by solving the following DLE*

$$\boldsymbol{A}_1 \boldsymbol{R}_{12} \boldsymbol{A}_2^{\mathrm{T}} - \boldsymbol{R}_{12} + \boldsymbol{H} = 0. \tag{15}$$

## 4.4 Constraint-Aware Updates

We cannot preserve the orthogonality of $V, U$ and the stability of $\Lambda$ if we use conventional gradient-descent methods to update the parameters $\Lambda, V, U$ of (12). Optimization on the space of orthogonal matrices is a well-studied problem [15]. Here, we employ the Cayley transformation [15] to maintain orthogonality for $V$ and $U$. In particular, we update $V$ by

$$V = V - \tau L_V (\mathbf{I}_{2n} + \frac{\tau}{2} R_V^{\mathrm{T}} L_V)^{-1} R_V^{\mathrm{T}} V, \tag{16}$$

where $L_V = [\nabla V, V]$ and $R_V = [V, -\nabla V]$, $\nabla V$ is the gradient of the objective w.r.t. $V$, and $\tau$ is the learning rate. Similarly, to update $U$, we use

$$U = U - \tau L_U (\mathbf{I}_{2n} + \frac{\tau}{2} R_U^{\mathrm{T}} L_U)^{-1} R_U^{\mathrm{T}} U, \tag{17}$$

where $L_U = [\nabla U, U]$ and $R_U = [U, -\nabla U]$. As shown in [15], the Cayley transform follows the descent curve, thus updating $V$ by Eq. (16) and $U$ by Eq. (17) decreases the objective for sufficiently small $\tau$.

To accomplish stability, we apply the following bound normalization on $\Lambda$, *i.e.*,

$$\lambda_k = \frac{\varepsilon}{\max(\varepsilon, |\lambda_k - \tau \nabla \lambda_k|)} (\lambda_k - \tau \nabla \lambda_k), \tag{18}$$

where $\lambda_k$ is the $k$-th diagonal element of $\Lambda$; $\nabla \lambda_k$ denotes the gradient w.r.t. $\lambda_k$; and $\varepsilon < 1$ is a threshold (we set $\varepsilon = 0.99$ in all of our experiments in this paper). From the above, we immediately have the following result,

**Theorem 5.** *The update direction in Eq. (18) is a descent direction.*

The authors in [5] constrain the eigenvalues of the transition matrix to be in $(-1, 1)$ using a Sigmoid function. However, the Sigmoid function is easier to saturate and its gradient will vanish when $\lambda_k$ is close to the bound. In contrast, Eq. (18) does not suffer from this issue.

For reader's convenience, all the aforementioned details for optimizing (11) are summarized in Algorithm 1. The full details about how to use Algorithm 1 to solve clustering and sparse coding are provided in the supplementary material.

---

**Algorithm 1** The PGD method to optimize problem (11)

---

**Input:** The given tuples $\{(A_i, C_j)\}$; the initialization of $(A, C)$; and the learning rate $\tau$;
According to Theorem 1, compute the canonical formulations of $\{(A_i, C_i)\}_{i=1}^N$ and $(A, C)$ as $\{(\Lambda_i, V_i, U_i)\}_{i=1}^N$ and $(\Lambda, V, U)$, respectively;
**for** $t = 1$ **to** maxIter **do**
    Compute the gradients according to Theorem 4: $\nabla \Lambda, \nabla V, \nabla U$;
    Update $V$: $V = V - \tau L_V (\mathbf{I}_{2n} + \frac{\tau}{2} R_V^{\mathrm{T}} L_V)^{-1} R_V^{\mathrm{T}} V$ with $L_V$ and $R_V$ defined in Eq. (16);
    Update $U$: $U = U - \tau L_U (\mathbf{I}_{2n} + \frac{\tau}{2} R_U^{\mathrm{T}} L_U)^{-1} R_U^{\mathrm{T}} U$ with $L_U$ and $R_U$ defined in Eq. (17);
    Update $\Lambda$: $\lambda_k = \frac{\varepsilon}{\max(\varepsilon, |\lambda_k - \tau \nabla \lambda_k|)} (\lambda_k - \tau \nabla \lambda_k)$;
**end for**
**Output:** the system tuple $(\Lambda, V, U)$.

---

## 4.5 Extensions for Other Kernels

The proposed solution is general in nature and can be used with other kernel functions such as the Martin kernel [25] and Binet-Cauchy kernel [19]. The Martin kernel is defined as

$$k_m\big((A_1, C_1), (A_2, C_2)\big) = \det\Big(G_{11}^{-1} G_{12} G_{22}^{-1} G_{21}\Big), \tag{19}$$

with $G_{ij}$ as in Eq.(5). The determinant version of the Binet-Cauchy kernel is defined as

$$k_b\big((A_1, C_1), (A_2, C_2)\big) = \det\Big(C_1 M C_2^{\mathrm{T}}\Big), \tag{20}$$

where $M$ satisfies $e^{-\lambda_b} A_1 M A_2^{\mathrm{T}} - M = -x_1(1) x_2^{\mathrm{T}}(1)$, $\lambda_b$ is the exponential discounting rate, and $x_1(1), x_2(1)$ are the initial hidden states of the two compared LDSs. Both the Martin kernel and Binet-Cauchy kernel are computed by DLEs. Thus, Theorem 4 can be employed to compute the gradients w.r.t. the system tuple for them.

# 5 Experiments

In this section, we first compare the performance of our proposed method (see Algorithm 1), called as PGD, with previous state-of-the-art methods for the task of clustering and sparse coding using the *DynTex++* [28] dataset. We then evaluate the classification accuracies of various state-of-the-art methods with PGD on two video datasets, namely the *YUPENN* [29] and the *DynTex* [30] datasets. The above datasets have been widely used in evaluating LDS-based algorithms in the literature, and their details are presented in the supplementary material. In all experiments, the hidden order of LDS ($n$ in Eq. (1)) is fixed to 10. To learn an LDS dictionary, we use the sparsity factor of 0.1 ($\lambda$ in Eq.(9)). The LDS tuples for all input sequences are learned by the method in [1] and the transition matrices are stabilized by the soft-normalization technique in [5].

## 5.1 Models Comparison

This experiment uses the *DynTex++* datasets. We extract the histogram of LBP from Three Orthogonal Planes (LBP-TOP) [31] by splitting each video into sub-videos of length 8, with a 6-frame overlap. The LBP-TOP features are fed to LDSs to identify the system parameters. For clustering, we compare our PGD with the MDS method with the Martin Kernel [11] and the Align algorithm [14]. For sparse coding, two related methods are compared: Grass [13] and LDSST [5]. We follow [13] and use 3-step observability matrices for the Grass method (hence Grass-3 below). In LDSST, the transition matrices are enforced to be symmetric. All algorithms are randomly initialized and the average results over 10 times are reported.

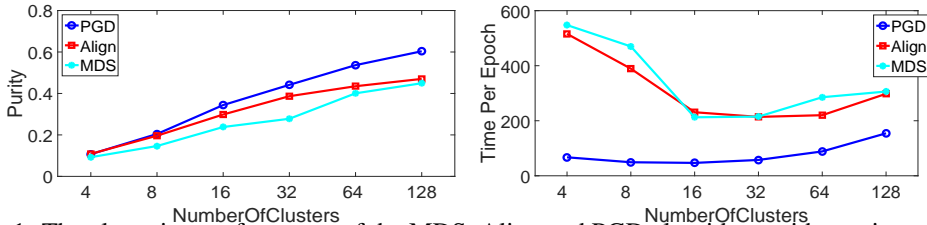

Figure 1: The clustering performance of the MDS, Align and PGD algorithms with varying number of clusters on *DynTex++*.

### 5.1.1 Clustering

To evaluate the clustering performance, we apply the *purity* metric [32], which is given by $p = \frac{1}{N}\sum_k \max_i c_{i,k}$, where $c_{i,k}$ counts the number of samples from $i$-th class in $k$-th cluster; $N$ is the number of the data. A higher purity means a better performance.

For the Align algorithm, we varied the learning rate when optimizing the aligning matrices and chose the value that delivered the best performance. For our PGD algorithm, we selected the learning rate as 0.1 for $\Lambda$ and $V$ and 1 for $U$. Fig. 1 reports the clustering performance of the compared methods. Our method consistently outperforms both MDS and Align methods over various number of clusters. We also report the running time for one epoch of each algorithm in Fig. 1. Here, one epoch means one update of the clustering centers through all data samples. Fig. 1 shows that PGD performs faster that both the MDS and Align algorithms, probably because the MDS method recomputes the kernel-matrix for the embedding at each epoch and the Align algorithm calculates the aligning distance in an iterative way.

### 5.1.2 Sparse Coding

In this experiment, we used half of samples from *DynTex++* for training the dictionary and the other half for testing. As the objective of (11) is in a sum-minimize form, we can employ the stochastic version of Algorithm 1 to optimize (11) for large-scale dataset. This can be achieved by sampling a mini-batch to update the system tuple at each iteration. Therefore, in addition to the full batch version, we also carried out the stochastic PGD with the mini-bach of size 128, which is denoted as PGD-128. The learning rates of both full PGD and PGD-128 were selected as 0.1 for $\Lambda$ and $V$ and 1 for $U$, and their values were decreased by half every 10 epoch. Different from PGD, the Grass and LDSST methods require the whole dataset in hand for learning the dictionary at each epoch, and thus they can not support the update via mini-batches.

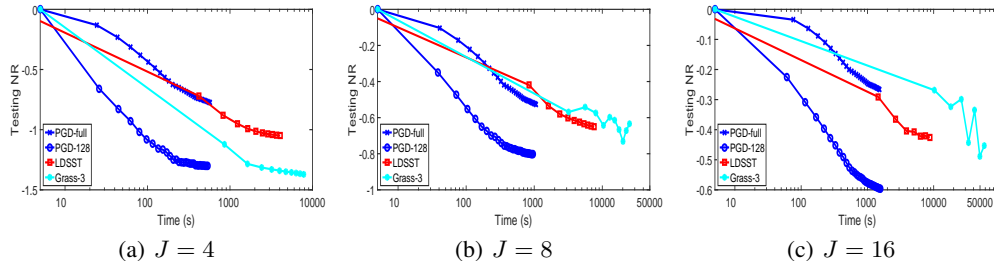

| (a) $J = 4$ | (b) $J = 8$ | (c) $J = 16$ |

Figure 2: Testing reconstruction errors of Grass-3, LDSST, PGD-full and PGD-128 with different dictionary sizes on *DynTex++*. The PGD-128 method converges much faster than other counterparts. Although Grass-3 converges to a bit smaller error than PGD-128 when $J = 4$ (see (a)), it performs worse than PGD-128 when the value of $J$ is increasing (see (b) and (c)).

It is unfair to directly compare the reconstruction errors (Eq. (9)) of different methods, since their values are calculated by different metrics. Therefore, we make use of the normalized reconstruction error defined as $NR = \frac{R_t - R_{init}}{R_{init}}$, where $R_{init}$ and $R_t$ are corresponded to the reconstruction errors at the initial step and the $t$-th epoch, respectively. Fig. 2 shows the normalized reconstruction errors on testing set of PGDs, Grass-3 and the LDSST method during the learning process for various dictionary sizes. PGD-128 converges to lower errors than PGD-full on all experiments, indicating that the stochastic sampling strategy is helpful to escaping from the poor local minima. PGD-128 consistently outperforms both Grass-3 and LDSST in terms of the learning speed and the final error.

The computational complexities of updating one dictionary atom for the Grass and the LDSST method are $O((J + N)L^2n^2m^2))$ and $O((J + N)n^2m^2))$, respectively. Here, $J$ is the dictionary size, $N$ is the number of data, and $n$ and $m$ are LDS parameters defined in Eq. (1). In contrast, PGD requires to calculate the projected gradients of the canonical tuples which scales to only $O((J + N)n^2m)$. As shown in Fig. 2, PGD is more than 50 times faster than the Grass-3 and LDSST methods per epoch.

## 5.2 Video Classification

Classifying *YUPENN* or *DynTex* videos is challenging as the videos are recoded under various viewpoints and scales. To deliver robust features, we implement two kinds of high-level pooling frameworks: Bag-of-Systems (BoS) [11] and Spatial-Temporal-Pyramid-Matching (STPM) [16][3]. In particular, 1) BoS is performed with the clustering methods, *i.e.*, MDS, Align and PGD. The BoS framework models the local spatio-temporal blocks with LDSs and then clusters the LDS descriptors to obtain the codewords; 2)The STPM framework works in conjunction with the sparse coding approaches (*i.e.*, Grass-3, LDSST and the PGD methods). Unlike BoS that represents a video by unordered local descriptors, STPM partitions a video into segments under different scales (2-level scales are considered here) and concatenates all local descriptors for each segment to form a vectorized representation. The codewords are provided by learning a dictionary. For the BoS methods, we apply the nonlinear SVM as the classifier where the radial basis kernel with $\chi^2$ distance [33] is employed; while for the STPM methods, we utilize linear SVM for classification.

Table 1: Mean classification accuracies (percentage) on the YUPENN and DynTex datasets.

| Datasets | References | +BoS | | | +STPM | | |
|---|---|---|---|---|---|---|---|
| | | MDS | Align | PGD | Grass-3 | LDSST | PGD |
| YUPENN | 85 [10] | 83.3 | 82.1 | **84.1** | 91.6 | 90.7 | **93.6** |
| DynTex | - | 59.5 | 62.7 | **65.4** | 75.1 | 75.1 | **76.5** |

**YUPENN.** The non-overlapping spatio-temporal blocks of size $8 \times 8 \times 25$ were sampled from the videos. The number of the codewords for all BoS and STPM methods was set to 128. We sampled 50 blocks from each video to learn the codewords for the MDS, Align, Grass-3 and LDSST methods. For PGD, we updated the codewords by mini-batches. To maintain the diversity within each mini-batch, a

hierarchical approach was used. In particular, at each iteration, we first randomly sampled 20 videos from the dataset and then sampled 4 blocks from each of the videos, leading to a mini-batch of size $N' = 80$. The learning rates were set as 0.5 for $\boldsymbol{\Lambda}$ and $\boldsymbol{V}$ and 5 for $\boldsymbol{U}$, and their values were decreased by half every 10 epochs. The test protocol is the leave-one-video-out as suggested in [29], leading to a total of 420 trials. Table 1 shows that the STPM methods achieve better accuracies than the BoS approaches; within the same pooling framework, our PGD always outperforms other compared models. For the probabilistic clustering method [10], the result on YUPENN is 85% reported in Table 1. Note that in [10], a richer number of dictionary has been applied.

**DynTex.** For the Dyntex dataset, the spatio-temporal blocks of size $16 \times 16 \times 50$ were sampled in a non-overlapping way. The number of the codewords for all methods was chosen as 64. We applied the same sampling strategy as that on *YUPENN* to learn the codewords for all compared methods. As shown in Table 1, the proposed method is superior compared to the studied models with both BoS and STPM coding strategies.

## 6    Conclusion

We propose an efficient Projected-Gradient-Decent (PGD) method to optimize problem (11). Our algorithm can be used to perform clustering and sparse coding with LDSs. In contrast to previous studies, our solution avoids any approximation in LDS modeling or during the optimization process. Extensive experiments on clustering and sparse coding verify the effectiveness of the proposed method in terms of the convergence performance and learning speed. We also explore the combination of PGD with two high-level pooling frameworks, namely Bag-of-Systems (BoS) and Spatial-Temporal-Pyramid-Matching for video classification. The experimental results demonstrate that our PGD method outperforms state-of-the-art methods consistently.

## Acknowledgments

This research was supported in part by the National Science Foundation of China (NSFC) (Grant No: 91420302, 91520201,61210013 and 61327809), the NSFC and the German Research of Foundation (DFG) in project Crossmodal Learning (Grant No: NSFC 61621136008/ DFG TRR-169), and the National High-Tech Research and Development Plan under Grant 2015AA042306. Besides, Tong Zhang was supported by Australian Research Council's Discovery Projects funding scheme (project DP150104645).

## Footnotes

[1]In general, $(\boldsymbol{A}, \boldsymbol{C}) \sim (\boldsymbol{P}^{-1}\boldsymbol{A}\boldsymbol{P}, \boldsymbol{C}\boldsymbol{P})$ for $\boldsymbol{P} \in \text{GL}(n)$ with $\text{GL}(n)$ denoting non-singular $n \times n$ matrices. Since we are interested in orthogonal measurement matrices (*i.e.*, $\boldsymbol{C} \in \text{ST}(m, n)$), the equivalent class takes the form described in Eq. (3).

[2]All the proofs of the theorems in this paper are provided in the supplementary material.

[3] In the experiments, we consider the projection kernel as defined in Eq. (5). We have also conducted additional experiments by considering a new kernel, namely the Martin kernel (Eq. (19)). The results are provided in the supplementary material.

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
