[Supplementary Material · appendix.pdf]

# Supplementary Material for
# Efficient Optimization for Linear Dynamical Systems with Applications to Clustering and Sparse Coding

**Wenbing Huang[1,3], Mehrtash Harandi[2], Tong Zhang[2]**

**Lijie Fan[3], Fuchun Sun[3], Junzhou Huang[1]**
[1] Tencent AI Lab. ;
[2] Data61, CSIRO and Australian National University, Australia;
[3] Department of Computer Science and Technology, Tsinghua University,
Tsinghua National Lab. for Information Science and Technology (TNList);
[1]{helendhuang, joehhuang}@tencent.com
[2]{mehrtash.harandi@data61.csiro.au, tong.zhang@anu.edu.cn}
[3]{flj14@mails, fcsun@mail}.tsinghua.edu.cn

This supplementary material provides the proofs of Theorems 1, 4 and 5, and presents the full flowcharts of applying PGD (Algorithm 1) for clustering and sparse coding. Moreover, we also introduce more details of the datasets *YUPENN* [1], *DynTex* [2] and *DynTex++* [3] that are applied in our experiments. Finally, we provide additional experimental evaluations to compare the classification accuracies between the projection and Martin kernels.

In this supplementary material, bold capital letters denote matrices (*e.g.*, $\boldsymbol{X}$) and bold lower-case letters denote column vectors (*e.g.*, $\boldsymbol{x}$). $\mathbf{I}_n$ is the $n \times n$ identity matrix. The orthogonal group is denoted by $\mathcal{O}(n)$, *i.e.*, $\mathcal{O}(n) = \{\boldsymbol{R} \in \mathbb{R}^{n \times n} | \boldsymbol{R}\boldsymbol{R}^T = \boldsymbol{R}^T\boldsymbol{R} = \mathbf{I}_n\}$. $\|\cdot\|_1$ is the $\ell_1$ norm of a vector; $\|\cdot\|_F$ is the Frobenius norm of a matrix. $\boldsymbol{X}^{\mathrm{T}}$ denotes the matrix transposition. $\boldsymbol{X}$ : returns the vectorized elements from the columns of $\boldsymbol{X}$. $\otimes$ denotes the Kronecker-product. $d\boldsymbol{X}$ performs the differential operation on $\boldsymbol{X}$.

## 1 Proofs

**Theorem 1.** *For any given LDS, the system tuple $(\boldsymbol{A}, \boldsymbol{C}) \in \mathbb{R}^{n \times n} \times \mathbb{R}^{m \times n}$ and all its equivalent representations have the canonical form $(\boldsymbol{\Lambda}\boldsymbol{V}, \boldsymbol{U})$, where $\boldsymbol{U} \in \mathrm{ST}(m, n)$, $\boldsymbol{V} \in \mathcal{O}(n)$ and $\boldsymbol{\Lambda} \in \mathbb{R}^{n \times n}$ is diagonal with the diagonal elements arranged in a descend order,* i.e. $\lambda_1 \geq \lambda_2 \geq \cdots \geq \lambda_n$.

*Proof.* Let the SVD of $\boldsymbol{A}$ be $\boldsymbol{U}_A \boldsymbol{\Lambda} \boldsymbol{V}_A^{\mathrm{T}}$. According to P-equivalence (Eq.(3)), we obtain

$$
\begin{aligned}
(\boldsymbol{A}, \boldsymbol{C}) &\sim (\boldsymbol{U}_A^{\mathrm{T}} \boldsymbol{A} \boldsymbol{U}_A, \boldsymbol{C}\boldsymbol{U}_A) \\
&\sim (\boldsymbol{\Lambda}\boldsymbol{V}_A^{\mathrm{T}} \boldsymbol{U}_A, \boldsymbol{C}\boldsymbol{U}_A) \\
&= (\boldsymbol{\Lambda}\boldsymbol{V}, \boldsymbol{U}),
\end{aligned}
\tag{21}
$$

where $\boldsymbol{V} = \boldsymbol{V}_A^{\mathrm{T}} \boldsymbol{U}_A$ and $\boldsymbol{U} = \boldsymbol{C}\boldsymbol{U}_A$ are orthogonal. $\qquad\square$

Several matrix computation properties are applied for the proof of Theorem 4:

1. $d(\boldsymbol{Y}\boldsymbol{Z}) := (\boldsymbol{I} \otimes \boldsymbol{Y})d\boldsymbol{Z} : + (\boldsymbol{Z}^{\mathrm{T}} \otimes \boldsymbol{I})d\boldsymbol{Y} :$
2. $(\boldsymbol{A}\boldsymbol{B}\boldsymbol{C}) := (\boldsymbol{C}^{\mathrm{T}} \otimes \boldsymbol{A})\boldsymbol{B} :$

3. $(\boldsymbol{ABC}) :^{\mathrm{T}} = \boldsymbol{B} :^{\mathrm{T}} (\boldsymbol{C} \otimes \boldsymbol{A}^{\mathrm{T}})$

4. $(\boldsymbol{A} \otimes \boldsymbol{B})^{\mathrm{T}} = (\boldsymbol{A}^{\mathrm{T}} \otimes \boldsymbol{B}^{\mathrm{T}})$

Interested readers can find more details in `http://www.ee.ic.ac.uk/hp/staff/dmb/matrix/calculus.html`.

For better readability, we repeat Theorem 4 before its proof.

**Theorem 4.** *Let the extended observability matrices of two LDSs $(\boldsymbol{A}_1, \boldsymbol{C}_1)$ and $(\boldsymbol{A}_2, \boldsymbol{C}_2)$ be $\boldsymbol{O}_1$ and $\boldsymbol{O}_2$, respectively. Furthermore, let $\boldsymbol{G}_{12} = \boldsymbol{O}_1^{\mathrm{T}} \boldsymbol{O}_2 = \sum_{t=0}^{\infty} (\boldsymbol{A}_1^{\mathrm{T}})^t \boldsymbol{C}_1^{\mathrm{T}} \boldsymbol{C}_2 \boldsymbol{A}_2^t$ be the product-matrix between $\boldsymbol{O}_1$ and $\boldsymbol{O}_2$. Given the gradient of the objective function with respect to the product-matrix $\frac{\partial \Gamma}{\partial \boldsymbol{G}_{12}} \doteq \boldsymbol{H}$, the gradients with respect to the system parameters are*

$$
\begin{aligned}
\frac{\partial \Gamma}{\partial \boldsymbol{A}_1} &= \boldsymbol{G}_{12} \boldsymbol{A}_2 \boldsymbol{R}_{12}^{\mathrm{T}}, \quad \frac{\partial \Gamma}{\partial \boldsymbol{C}_1} = \boldsymbol{C}_2 \boldsymbol{R}_{12}^{\mathrm{T}}, \\
\frac{\partial \Gamma}{\partial \boldsymbol{A}_2} &= \boldsymbol{G}_{12}^{\mathrm{T}} \boldsymbol{A}_1 \boldsymbol{R}_{12}, \quad \frac{\partial \Gamma}{\partial \boldsymbol{C}_2} = \boldsymbol{C}_1 \boldsymbol{R}_{12},
\end{aligned}
\tag{22}
$$

*where $\boldsymbol{R}_{12}$ is obtained by solving the following DLE*

$$
\boldsymbol{A}_1 \boldsymbol{R}_{12} \boldsymbol{A}_2^{\mathrm{T}} - \boldsymbol{R}_{12} + \boldsymbol{H} = 0.
\tag{23}
$$

*Proof.* The definition $\boldsymbol{G}_{12} = \boldsymbol{O}_1^{\mathrm{T}} \boldsymbol{O}_2$ implies that

$$
\boldsymbol{A}_1^{\mathrm{T}} \boldsymbol{G}_{12} \boldsymbol{A}_2 - \boldsymbol{G}_{12} = -\boldsymbol{C}_1^{\mathrm{T}} \boldsymbol{C}_2.
\tag{24}
$$

By vectorizing and computing the differential on both sides of Eq. (24), we arrive at

$$
\begin{aligned}
& \boldsymbol{A}_2^{\mathrm{T}} \boldsymbol{G}_{12}^{\mathrm{T}} \otimes \boldsymbol{I}_n d\boldsymbol{A}_1^{\mathrm{T}} : + \boldsymbol{I}_n \otimes \boldsymbol{A}_1^{\mathrm{T}} \boldsymbol{G}_{12} d\boldsymbol{A}_2 : + \boldsymbol{C}_2^{\mathrm{T}} \otimes \boldsymbol{I}_n d\boldsymbol{C}_1^{\mathrm{T}} : + \boldsymbol{I}_n \otimes \boldsymbol{C}_1^{\mathrm{T}} d\boldsymbol{C}_2 : \\
=\ & (\boldsymbol{I}_{n^2} - \boldsymbol{A}_2^{\mathrm{T}} \otimes \boldsymbol{A}_1^{\mathrm{T}}) d\boldsymbol{G}_{12} :
\end{aligned}
\tag{25}
$$

Thus,

$$
\frac{\partial \boldsymbol{G}_{12} :}{\partial \boldsymbol{A}_1^{\mathrm{T}} :} = (\boldsymbol{I}_{n^2} - \boldsymbol{A}_2^{\mathrm{T}} \otimes \boldsymbol{A}_1^{\mathrm{T}})^{-1} (\boldsymbol{A}_2^{\mathrm{T}} \boldsymbol{G}_{12}^{\mathrm{T}} \otimes \boldsymbol{I}_n),
\tag{26}
$$

$$
\frac{\partial \boldsymbol{G}_{12} :}{\partial \boldsymbol{A}_2 :} = (\boldsymbol{I}_{n^2} - \boldsymbol{A}_2^{\mathrm{T}} \otimes \boldsymbol{A}_1^{\mathrm{T}})^{-1} (\boldsymbol{I}_n \otimes \boldsymbol{A}_1^{\mathrm{T}} \boldsymbol{G}_{12}),
\tag{27}
$$

$$
\frac{\partial \boldsymbol{G}_{12} :}{\partial \boldsymbol{C}_1^{\mathrm{T}} :} = (\boldsymbol{I}_{n^2} - \boldsymbol{A}_2^{\mathrm{T}} \otimes \boldsymbol{A}_1^{\mathrm{T}})^{-1} (\boldsymbol{C}_2^{\mathrm{T}} \otimes \boldsymbol{I}_n),
\tag{28}
$$

$$
\frac{\partial \boldsymbol{G}_{12} :}{\partial \boldsymbol{C}_2 :} = (\boldsymbol{I}_{n^2} - \boldsymbol{A}_2^{\mathrm{T}} \otimes \boldsymbol{A}_1^{\mathrm{T}})^{-1} (\boldsymbol{I}_n \otimes \boldsymbol{C}_1^{\mathrm{T}}),
\tag{29}
$$

where the invertibility of the term $(\boldsymbol{I}_{n^2} - \boldsymbol{A}_2^{\mathrm{T}} \otimes \boldsymbol{A}_1^{\mathrm{T}})$ is guaranteed by the stability of $\boldsymbol{A}_1$ and $\boldsymbol{A}_2$.

Under the vectorized form, we have $\frac{\partial \Gamma}{\partial \boldsymbol{G}_{12:}} = (\boldsymbol{G}_{12} :)^{\mathrm{T}}$. Applying the chain rule, we obtain

$$
\frac{\partial \Gamma}{\partial \boldsymbol{A}_1^{\mathrm{T}} :} = \frac{\partial \Gamma}{\partial \boldsymbol{G}_{12} :} \frac{\partial \boldsymbol{G}_{12} :}{\partial \boldsymbol{A}_1^{\mathrm{T}} :} = (\boldsymbol{R}_{12} :)^{\mathrm{T}} (\boldsymbol{A}_2^{\mathrm{T}} \boldsymbol{G}_{12}^{\mathrm{T}} \otimes \boldsymbol{I}_n) = (\boldsymbol{R}_{12} \boldsymbol{A}_2^{\mathrm{T}} \boldsymbol{G}_{12}^{\mathrm{T}} :)^{\mathrm{T}},
\tag{30}
$$

$$
\frac{\partial \Gamma}{\partial \boldsymbol{A}_2 :} = \frac{\partial \Gamma}{\partial \boldsymbol{G}_{12} :} \frac{\partial \boldsymbol{G}_{12} :}{\partial \boldsymbol{A}_2 :} = (\boldsymbol{R}_{12} :)^{\mathrm{T}} (\boldsymbol{I}_n \otimes \boldsymbol{A}_1^{\mathrm{T}} \boldsymbol{G}_{12}) = (\boldsymbol{G}_{12}^{\mathrm{T}} \boldsymbol{A}_1 \boldsymbol{R}_{12} :)^{\mathrm{T}},
\tag{31}
$$

$$
\frac{\partial \Gamma}{\partial \boldsymbol{C}_1^{\mathrm{T}} :} = \frac{\partial \Gamma}{\partial \boldsymbol{G}_{12} :} \frac{\partial \boldsymbol{G}_{12} :}{\partial \boldsymbol{C}_1^{\mathrm{T}} :} = (\boldsymbol{R}_{12} :)^{\mathrm{T}} (\boldsymbol{C}_2^{\mathrm{T}} \otimes \boldsymbol{I}_n) = (\boldsymbol{R}_{12} \boldsymbol{C}_2^{\mathrm{T}} :)^{\mathrm{T}},
\tag{32}
$$

$$
\frac{\partial \Gamma}{\partial \boldsymbol{C}_2 :} = \frac{\partial \Gamma}{\partial \boldsymbol{G}_{12} :} \frac{\partial \boldsymbol{G}_{12} :}{\partial \boldsymbol{C}_2 :} = (\boldsymbol{R}_{12} :)^{\mathrm{T}} (\boldsymbol{I}_n \otimes \boldsymbol{C}_1^{\mathrm{T}}) = (\boldsymbol{C}_1 \boldsymbol{R}_{12} :)^{\mathrm{T}},
\tag{33}
$$

where we have defined

$$
(\boldsymbol{R}_{12} :)^{\mathrm{T}} = (\boldsymbol{H} :)^{\mathrm{T}} (\boldsymbol{I}_{n^2} - \boldsymbol{A}_2^{\mathrm{T}} \otimes \boldsymbol{A}_1^{\mathrm{T}})^{-1}.
\tag{34}
$$

Then,

$$
\begin{aligned}
& (\boldsymbol{R}_{12}:)^{\mathrm{T}} = (\boldsymbol{H}:)^{\mathrm{T}}(\boldsymbol{I}_{n^2} - \boldsymbol{A}_2^{\mathrm{T}} \otimes \boldsymbol{A}_1^{\mathrm{T}})^{-1} \\
\Rightarrow \quad & (\boldsymbol{R}_{12}:)^{\mathrm{T}}(\boldsymbol{I}_{n^2} - \boldsymbol{A}_2^{\mathrm{T}} \otimes \boldsymbol{A}_1^{\mathrm{T}}) = (\boldsymbol{H}:)^{\mathrm{T}} \\
\Rightarrow \quad & (\boldsymbol{I}_{n^2} - \boldsymbol{A}_2 \otimes \boldsymbol{A}_1)(\boldsymbol{R}_{12}:) = \boldsymbol{H}: \\
\Rightarrow \quad & \boldsymbol{R}_{12}: -(\boldsymbol{A}_2 \otimes \boldsymbol{A}_1)\boldsymbol{R}_{12} := \boldsymbol{H}: \\
\Rightarrow \quad & \boldsymbol{R}_{12}: -(\boldsymbol{A}_1 \boldsymbol{R}_{12} \boldsymbol{A}_2^{\mathrm{T}}) := \boldsymbol{G}_{12}: \\
\Rightarrow \quad & \boldsymbol{R}_{12} - \boldsymbol{A}_1 \boldsymbol{R}_{12} \boldsymbol{A}_2^{\mathrm{T}} = \boldsymbol{H} \\
\Rightarrow \quad & \boldsymbol{A}_1 \boldsymbol{R}_{12} \boldsymbol{A}_2^{\mathrm{T}} - \boldsymbol{R}_{12} + \boldsymbol{H} = 0
\end{aligned}
\tag{35}
$$

Substituting the matrix $\boldsymbol{R}_{12}$ into Eq. (30-33) concludes the proof.

Note that one can directly derive $\boldsymbol{R}_{12}$ from Eq. (34). However, it will increase the computational complexity drastically as the inversion of $(\boldsymbol{I}_{n^2} - \boldsymbol{A}_2^{\mathrm{T}} \otimes \boldsymbol{A}_1^{\mathrm{T}})^{-1}$ leads to $O(n^6)$ flops. We recall that our solution by using the DLE (Eq. (35)) only requires $O(n^3)$ flops. □

**Theorem 5.** *The update direction in Eq.(18) is a descent direction.*

*Proof.* We denote the update in Eq.(18) by $d$. Then $d = \frac{\varepsilon}{\max(\varepsilon, |\lambda_k - \tau \nabla \lambda_k|)}(\lambda_k - \tau \nabla \lambda_k) - \lambda_k$. To prove $d$ is along a descent direction, we need to prove $d^{\mathrm{T}} \nabla \lambda_k < 0$ for small $\tau$. To be specific, $d^{\mathrm{T}} \nabla \lambda_k = -a\tau (\nabla \lambda_k)^{\mathrm{T}} \nabla \lambda_k - (1-a)\lambda_k^{\mathrm{T}} \nabla \lambda_k$, where $0 < a = \frac{\varepsilon}{\max(\varepsilon, |\lambda_k - \tau \nabla \lambda_k|)} \leq 1$.

Thus,

$$
\begin{aligned}
d^{\mathrm{T}} \nabla \lambda_k & \leq -a\tau |\nabla \lambda_k|^2 + (1-a)|\lambda_k^{\mathrm{T}} \nabla \lambda_k|, \\
& \leq -a\tau |\nabla \lambda_k|^2 + (1-a)|\lambda_k||\nabla \lambda_k|, \quad \text{(Cauchy Schwarz inequality)} \\
& \leq -a\tau |\nabla \lambda_k|^2 + (1-a)\varepsilon |\nabla \lambda_k|, \\
& = a(-\tau |\nabla \lambda_k|^2 + (\frac{1}{a} - 1)\varepsilon |\nabla \lambda_k|), \\
& = a(-\tau |\nabla \lambda_k|^2 + (\max(\varepsilon, |\lambda_k - \tau \nabla \lambda_k|) - \varepsilon)|\nabla \lambda_k|), \\
& \leq a(-\tau |\nabla \lambda_k|^2 + (\max(\varepsilon, |\lambda_k| + \tau |\nabla \lambda_k|) - \varepsilon)|\nabla \lambda_k|), \\
& \leq a(-\tau |\nabla \lambda_k|^2 + \tau |\nabla \lambda_k||\nabla \lambda_k|), \\
& = 0,
\end{aligned}
\tag{36}
$$

where $d^{\mathrm{T}} \nabla \lambda_k = 0$ if and only if $|\lambda_k| = \varepsilon$, $\lambda_k$ and $\nabla \lambda_k$ have the opposite directions. □

## 2 Gradients of the kernels with respect to the LDS parameters

For the reader's convenience, we also provide the gradients of the projection kernel (Eq. (5)) and the Martin kernel (Eq. (19)). These gradients are necessary to pass the gradients from the loss back to the LDS parameters.

### 2.1 Projection kernel

Suppose we are updating the $r$-th dictionary atom and passing the gradient through the kernel between $\boldsymbol{D}_r$ and $\boldsymbol{D}_j$. Recall that the projection kernel is given by

$$
k(\boldsymbol{D}_r, \boldsymbol{D}_j) = \mathrm{Tr}\left(\boldsymbol{G}_{rr}^{-1} \boldsymbol{G}_{rj} \boldsymbol{G}_{jj}^{-1} \boldsymbol{G}_{jr}\right).
$$

Then,

$$
\frac{\partial k(\boldsymbol{D}_r, \boldsymbol{D}_j)}{\partial \boldsymbol{G}_{rr}} = -\boldsymbol{G}_{rr}^{-1} \boldsymbol{G}_{rj} \boldsymbol{G}_{jj}^{-1} \boldsymbol{G}_{jr} \boldsymbol{G}_{rr}^{-1},
\tag{37}
$$

$$
\frac{\partial k(\boldsymbol{D}_r, \boldsymbol{D}_j)}{\partial \boldsymbol{G}_{rj}} = \boldsymbol{G}_{rr}^{-1} \boldsymbol{G}_{rj} \boldsymbol{G}_{jj}^{-1},
\tag{38}
$$

$$
\frac{\partial k(\boldsymbol{D}_r, \boldsymbol{D}_j)}{\partial \boldsymbol{G}_{jr}} = \boldsymbol{G}_{jj}^{-1} \boldsymbol{G}_{jr} \boldsymbol{G}_{rr}^{-1}.
\tag{39}
$$

## 2.2 Martin kernel

The Martin kernel is defined as

$$k(\boldsymbol{D}_r, \boldsymbol{D}_j) \;=\; \det\left(\boldsymbol{G}_{rr}^{-1}\boldsymbol{G}_{rj}\boldsymbol{G}_{jj}^{-1}\boldsymbol{G}_{jr}\right).$$

Thus,

$$\frac{\partial k(\boldsymbol{D}_r, \boldsymbol{D}_j)}{\partial \boldsymbol{G}_{rr}} \;=\; -\det\left(\boldsymbol{G}_{rr}^{-1}\boldsymbol{G}_{rj}\boldsymbol{G}_{jj}^{-1}\boldsymbol{G}_{jr}\right)\boldsymbol{G}_{rr}^{-1}, \tag{40}$$

$$\frac{\partial k(\boldsymbol{D}_r, \boldsymbol{D}_j)}{\partial \boldsymbol{G}_{rj}} \;=\; \det\left(\boldsymbol{G}_{rr}^{-1}\boldsymbol{G}_{rj}\boldsymbol{G}_{jj}^{-1}\boldsymbol{G}_{jr}\right)\boldsymbol{G}_{rj}^{-1}, \tag{41}$$

$$\frac{\partial k(\boldsymbol{D}_r, \boldsymbol{D}_j)}{\partial \boldsymbol{G}_{jr}} \;=\; \det\left(\boldsymbol{G}_{rr}^{-1}\boldsymbol{G}_{rj}\boldsymbol{G}_{jj}^{-1}\boldsymbol{G}_{jr}\right)\boldsymbol{G}_{jr}^{-1}. \tag{42}$$

## 3 Algorithms for clustering and sparse coding

In the paper, § 4 has demonstrated how to apply the PGD method to compute the mean for clustering and learn the dictionary atoms for sparse coding. We now embed the PGD method into the implementations of these two tasks and provide full details in Algorithms 2 and 3 below.

---

**Algorithm 2** The PGD method for clustering

---

**Input:** The data tuples $\{(\boldsymbol{A}_i, \boldsymbol{C}_i)\}_{i=1}^{N}$; the initialization of the means $\{(\boldsymbol{A}_{m_i}, \boldsymbol{C}_{m_i})\}_i^{C}$;
According to Theorem 1, compute the canonical formulations of $\{(\boldsymbol{A}_i, \boldsymbol{C}_i)\}_{i=1}^{N}$ and $\{(\boldsymbol{A}_{m_i}, \boldsymbol{C}_{m_i})\}_i^{C}$ as $\{(\boldsymbol{\Lambda}_i, \boldsymbol{V}_i, \boldsymbol{U}_i)\}_{i=1}^{N}$ and $\{(\boldsymbol{\Lambda}_{m_i}, \boldsymbol{V}_{m_i}, \boldsymbol{U}_{m_i})\}_{i=1}^{C}$, respectively;
**for** $t = 1$ **to** maxIter **do**
    Assign the data tuples to the closest clusters according to the given metric;
    **for** $i = 1$ **to** $C$ **do**
        Update the mean tuple $(\boldsymbol{\Lambda}_{m_i}, \boldsymbol{V}_{m_i}, \boldsymbol{U}_{m_i})$ of the $i$-th cluster via Algorithm 1;
    **end for**
**end for**
**Output:** the means $\{(\boldsymbol{A}_{m_i}, \boldsymbol{C}_{m_i})\}_i^{C}$.

---

---

**Algorithm 3** The PGD method for sparse coding

---

**Input:** The data tuples $\{(\boldsymbol{A}_i, \boldsymbol{C}_i)\}_{i=1}^{N}$; the initialization of the dictionary atoms $\{(\boldsymbol{A'}_j, \boldsymbol{C'}_j)\}_j^{J}$;
According to Theorem 1, compute the canonical formulations of $\{(\boldsymbol{A}_i, \boldsymbol{C}_i)\}_{i=1}^{N}$ and $\{(\boldsymbol{A}_{m_i}, \boldsymbol{C}_{m_i})\}_i^{C}$ as $\{(\boldsymbol{\Lambda}_i, \boldsymbol{V}_i, \boldsymbol{U}_i)\}_{i=1}^{N}$ and $\{(\boldsymbol{\Lambda'}_j, \boldsymbol{V'}_j, \boldsymbol{U'}_j)\}_{j=1}^{J}$, respectively;
**for** $t = 1$ **to** maxIter **do**
    Compute the sparse codes $\boldsymbol{z}_i$ given LDS dictionary by the homotopy-LARS algorithm [4];
    **for** $r = 1$ **to** $J$ **do**
        Update the $r$-th atom via Algorithm 1 with only one iteration;
    **end for**
**end for**
**Output:** the dictionary atoms $\{(\boldsymbol{A'}_j, \boldsymbol{C'}_j)\}_j^{J}$.

---

## 4 Datasets

YUPENN dataset introduced in [1] consists of fourteen dynamic scene categories where each category has 30 color videos. Analysing this dataset is challenging as the videos are obtained from various sources, *e.g.*, YouTube, BBC Motion Gallery and Getty Images. The videos have an average dimension of $250 \times 370 \times 145$ and vary significantly in resolution, frame rate, scene appearance, scale, illumination condition, and camera viewpoint. Representative examples from different classes are illustrated in Figure 1. We convert all videos to gray-scales and down-sample each frame to have a maximum spatial dimension of 128 pixels while keeping the original aspect ratio.

Figure 1: Examples of the YUPENN dataset demonstrating various dynamic scenes (*e.g*., beach, elevator and forest fire).

Figure 2: Examples from the DynTex dataset.

The *DynTex* dataset [2] contains $352 \times 288 \times 250$ videos recorded under different environmental conditions, scales and rotations (as illustrated in Figure 2). Three subsets, *i.e.*, *Alpha*, *Beta* and *Gamma* have been applied for classification benchmark in previous studies [2, 5]. However, both *Alpha* and *Beta* have a small number of videos, *i.e.*, 60 and 162, respectively; performing evaluations on them could be easily bias. Hence, we formulate a new dataset by combining the samples of the three subsets, leading to a larger dataset containing 307 videos of 12 classes: *Calm water*, *Escalator*, *Flags*, *Rotation*, *Sea*, *Smoke*, *Traffic*, *Fountain*, *Naked trees*, *Foliage*, *Grass* and *Flowers*. In particular, we first combine the classes *Trees* from *Alpha* and *Beta*, *Naked trees* and *Foliage* from *Gamma* into two non-overlap categories *Naked trees* and *Foliage*; and then combine the classes *Grass* from *Alpha*, *Vegetation* from *Beta*, and *Flowers* and *Grass* from *Gamma* into two categories *Flowers* and *Grass*; and finally integrate other classes of the three subsets. We resize all the videos to $128 \times 128$ of gray scale.

*DynTex++* [3] is an variant of *DynTex*, where the samples are extracted from the local regions of the videos in *DynTex*. It consists of 3600 videos of 36 classes with 100 videos of size $50 \times 50 \times 50$ per class.

## 5   Experimental comparison between different kernels

We have performed the classification experiments of PGD based on the projection kernel in the paper (Section 5.2). Now we provide additional experimental evaluations of PGD based on the Martin kernel. In practice, we find that the original definition of Martin kernel in Eq.(19) produces a very small value due to the determinant calculation. We thus revise the Martin kernel by computing powers of the original kernel, namely,

$$k_m\big((\boldsymbol{A}_1, \boldsymbol{C}_1), (\boldsymbol{A}_2, \boldsymbol{C}_2)\big) = \Big(\det\Big(\boldsymbol{G}_{11}^{-1}\boldsymbol{G}_{12}\boldsymbol{G}_{22}^{-1}\boldsymbol{G}_{21}\Big)\Big)^{\frac{1}{\gamma}}, \tag{43}$$

where $\gamma$ is set to be 100. We follow the same experimental set-ups as those in the paper. The classification accuracies of the projection kernel and Martin kernel are provided in Table 1. It is observed that Martin kernel almost works better in conjunction with PGD compared with the projection kernel.

Table 1: Mean classification accuracies of the projection and Martin kernels.

| Datasets | +BoS | | +STPM | |
|---|---|---|---|---|
| | Projection | Martin | Projection | Martin |
| YUPENN | 84.1 | **86.1** | 93.6 | **94.2** |
| DynTex | 65.4 | **66.7** | **76.5** | 74.5 |