[Reviews · NeurIPS 2017]

Reviewer 1



A very interesting paper that solves clustering and sparse approximation problems of LDSs by using a kernel-based formulation (kernel defined over pairs of LDSs defined using extended observability subspaces) and a projected gradient descent method. The central contribution of the paper, summarized in Theorem 4, is the calculation of the gradient terms by solving a discrete Lyapunov equation. Overall, significant improvements in accuracy and speedups are demonstrated for clustering and sparse coding and their application for video classification of dynamic textures. Strengths + Well-written paper + Strong algorithmic core with theoretical justification + Limited but sufficient evaluation Weaknesses - Section 4 is very tersely written (maybe due to limitations in space) and could have benefitted with a slower development for an easier read. - Issues of convergence, especially when applying gradient descent over a non-Euclidean space, is not addressed In all, a rather thorough paper that derives an efficient way to compute gradients for optimization on LDSs modeled using extended subspaces and kernel-based similarity. At one hand, this leads to improvements over some competing methods. Yet, at its core, the paper avoids handling of the harder topics including convergence and any analysis of the proposed optimization scheme. None the less, the derivation of the gradient computations is interesting by itself. Hence, my recommendation.

Reviewer 2



Summary: the paper proposes a clustering method for linear dynamical systems, based on minimizing the kernalized norm between extended observability spaces. Since the objective function contains terms involving discrete Lypanunov equations (DLEs), the paper derives how to pass the derivatives through, yielding a gradient descent algorithm. Experiments are presented on 2 datasets, for LDS clustering and sparse codebook learning. Originality: +1) novelty: the paper is moderately novel, but fairly straightforward. The difference with other related works [3,9,10,11,12] in terms of objective functions, assumptions/constraints, canonical forms should be discussed more. Quality: There are a few potential problems in the derivations and motivation. -1) Theorem 1: The canonical form should have an additional ineqaultiy constraint on Lambda, which is: lambda_1 > lambda_2 > lambda_3 > ... lambda_n which comes from Lambda being a matrix of the singular values of A. Imposing this constraint removes equivalent representations due to permutations of the state vector. This inequality constraint could also be included in (12) to truly enforce the canonical form. However, I think it is not necessary (see next point). -2) It's unclear why we actually need the canonical form to solve (10). Since gradient descent is used, we only care that we obtain a feasible solution, and the actual solution that we converge to will depend on the initial starting point. I suppose that using the canonical form allows for an easier way to handle the stability constraint of A, rho(A)<1, but this needs to be discussed more, and compared with the straightforward approach of directly optimizing (A,C). -3) Theorem 4: I'm not entirely convinced by the proof. In the supplemental, Line 35 shows an explicit equation for the vectorized R_12. Then why in (34) does it need to be transformed into a DLE? There is something not quite correct about the vectorization and differentiation step. Furthermore, it seems that the gradients could be directly computed from (24). For example, taking the derivative wrt A1: d/dA1 (A1^T G12 A2 - G12) = d/dA1 (-C1^T C2) G12 A2 + A1^T (dG12/dA1) A2 - (dG12/dA1) = 0 Hence dG12/dA1 can be obtained directly as a DLE. -4) Given that there are previous works on LDS clustering, experiments are somewhat lacking: - Since a kernel method is used, it would be interesting to see the results for directly using the linear kernel, or the other kernels in (19,20). - In the video classification experiment, there is no comparison to [3,8,11], which also perform LDS clustering. - No comparison with probabilistic clustering methods [21,22]. Clarity: - L210 - Theorem 4 cannot be "directly employed" on new kernels, since the theorem is specific to the kernel in (5). Some extension, or partial re-derivation is needed. - Other minor typos: - Eq 8: it should be mentioned why the k(S_m,S_m) term can be dropped. - Eq 9: the minimization is missing z_i. - Eq 10: the text is missing how (9) is solved. I'm assuming there are two sub-problems: sparse code estmiation, and codebook estimation with z_i fixed. (10) corresponds to the second step. Significance: Overall, the approach is reasonable and offers a solution to the problem of clustering LDS. However, more could have been done in terms of relating the work with previous similar methods, exploring other kernel functions, and more thorough comparison in the expeirments. Some steps in the derivation need to be clarified.